# Optimization of Mature Embryo-Based Tissue Culture and *Agrobacterium*-Mediated Transformation in Model Grass *Brachypodium distachyon*

**DOI:** 10.3390/ijms20215448

**Published:** 2019-10-31

**Authors:** Guangrun Yu, Jianyong Wang, Li Miao, Mengli Xi, Qiongli Wang, Kai Wang

**Affiliations:** 1Key Laboratory of Genetics, Breeding and Multiple Utilization of Crops, Ministry of Education, Fujian Provincial Key Laboratory of Haixia Applied Plant Systems Biology, Center for Genomics and Biotechnology, Fujian Agriculture and Forestry University, Fuzhou 350002, Fujian, China; 2Co-Innovation Center for Sustainable Forestry in Southern China/Key Laboratory of Forest Genetics and Biotechnology of Ministry of Education, Nanjing Forestry University, Nanjing 210037, Jiangsu, China; 3National Engineering Research Center of Sugarcane, Fujian Agriculture and Forestry University, Fuzhou 350002, Fujian, China

**Keywords:** *Brachypodium distachyon*, genetic transformation, mature embryo, seed shearing, photoautotrophic rooting

## Abstract

*Agrobacterium*-mediated genetic transformation is well established in the model grass *Brachypodium distachyon*. However, most protocols employ immature embryos because of their better regenerative capacity. A major problem associated with the immature embryo system is that they are available only during a limited time window of growing plants. In this study, we have developed an optimized *Agrobacterium*-mediated genetic transformation protocol that utilizes mature embryos. We have adopted seed shearing and photoautotrophic rooting (PR) in callus induction and root regeneration, respectively, with evident significant improvement in these aspects. We have also revealed that the newly developed chemical inducer Fipexide (FPX) had the ability to induce callus, shoots, and roots. By comparison, we have demonstrated that FPX shows higher efficiency in shoot generation than other frequently used chemicals in our mature embryo-based system. In addition, we demonstrated that the age of embryogenetic callus severely affects the transformation efficiency (TE), with the seven-week-old embryogenetic callus having the highest TE reaching 52.6%, which is comparable with that in immature embryo transformation. The new methodologies reported here will advance the development and utilization of *Brachypodium* as a new model system for grass genomics.

## 1. Introduction

*Brachypodium distachyon* (L.), as an important member of the family *Poaceae*, has been proposed by Draper et al. as a new model system for temperate grass research [1]. The species has the typical characteristics of model organisms including *Arabidopsis thaliana* and *Oryza sativa*, such as small genome (~272 Mbp), self-fertility, simple growth requirements, diploid in some ecotypes (2*n* = 10), rapid annual cycling, and high transformation efficiency (TE) [2,3,4]. Therefore, *B. distachyon* has been applied in the studies involving plant development, biotic or abiotic stresses, evolutionary, and system biology [5,6,7,8]. More importantly, *B. distachyon* exhibits a close phylogenetic relationship with important temperate cereals, such as wheat, oat, and barley [9] but has a comparatively simple genome [10].

The major concern in agricultural studies is how to utilize modern bio-techniques to improve the agronomic traits of crops. *Agrobacterium*-mediated genetic transformation is one of the most widely used techniques for transferring foreign genes of interest to the plant genome in modern crop breeding. Currently, *Agrobacterium*-mediated genetic transformation has become the baseline in both plant research and breeding and has been well established in a wide range of plants [11,12,13]. In *Brachypodium*, the *Agrobacterium*-mediated genetic transformation system was developed 20 years ago by Draper et al. [1]. Since then enormous efforts have been made toward improving TE by optimizing the regeneration and transformation performing in *B. distachyon*. Although both mature [2,14] and immature embryos [3,9,15,16,17] were used to obtain embryogenic callus formation, immature embryos are used most frequently in *Brachypodium* due to its better regeneration capacity [18]. However, an immature embryo is only available from growing plants during a certain time window. In contrast, mature embryos are available throughout the year and thus offer an attractive alternative to researchers working in the field of plant tissue culture. 

In this study, we have described an optimized mature embryo-based *Agrobacterium*-mediated transformation protocol in *Brachypodium* by introducing the seed shearing and photoautotrophic rooting (PR) in callus formation and root regeneration, respectively. We have also revealed that the new chemical inducer Fipexide (FPX) has higher efficiency in shoot regeneration as compared to the routinely used chemicals. In addition, the embryogenic calli with different ages were examined to achieve ~50% TE.

## 2. Results

### 2.1. Shearing Promotes Callus Formation by Breaking Dormancy of Bd21 Mature Embryo

Dormancy in *B. distachyon* seed is an inherent nature that can delay the new seed germination time. Previous studies indicate that a wound on seed endosperm can break the dormancy and shorten seed germination time in plants [19,20]. We, thus, hypothesize that damage on endosperm may also shorten or improve the callus induction. To validate our hypothesis, we performed a comparative analysis of callus induction between sheared and unsheared seeds as control. The seeds were cut along the embryo to remove the endosperm as completely as possible (Figure 1a). The seeds were then (10 seeds for each petri dish, three replications) cultured on the callus induction medium (CIM) with 2,4-Dichlorophenoxyacetic acid (2,4-D). As shown in Figure 1, the calli from sheared seeds were visible as early as day two (Figure 1b). Strikingly, small calli were detected from at least 80% (Callus induction rate = the number of induced calli/total number of seeds ×100%) sheared seeds at day three (Figure 1b), when the callus formation had just begun in control seeds (10–20%) (Figure 1c). At day nine, larger calli buds from sheared seeds were clearly visible (Figure 1b) while small calli patches were merely detected from 80% seeds in control (Figure 1c). In addition, callus numbers and status from sheared seeds were also highly consistent, while the control seeds showed high diversity in callus number and status (Figure 1), indicating the synchronous initiation and growth of callus from sheared seeds.

### 2.2. FPX Improves the Induction of Bd21 Shoot Regeneration

Recently, Nakano et al. [21] demonstrated that FPX could function as a chemical inducer that promotes callus formation and shoot regeneration in plants. To examine if FPX can improve callus induction in our system, we conducted a comparative analysis between FPX and the regularly used chemical 2,4-D. Our results revealed that callus were indeed induced on the FPX medium. We also analyzed other comparisons of the amount and status of calli. However, we did not obtain any significant difference (G.Y. and K.W. A robust tissue regeneration system in *Brachypodium distachyon*. Unpublished). 

We then examined the efficiency of FPX on shoot regeneration. A total of 300 Bd21 calli were cultured on shoot induction medium (SIM) containing Kinetin (KT), Thidiazuron (TDZ) and FPX, respectively. By comparing the numbers of regenerated shoots after one week, clear differences in shoot regeneration were obtained. Among them, FPX showed the highest regeneration rate at 21%, while TDZ and KT had significantly lower (*p* < 0.01) values at 7% and 4.3%, respectively (Figure 2). After two weeks, an increase in shoot regeneration rates for all treatments was observed (Figure 2). However, FPX still showed relatively higher induction efficiency (40.7%) than both KT (34%) and TDZ (25%) (Figure 2), indicating that FPX can improve the shoot formation in our mature seed system.

### 2.3. Photoautotrophic Rooting is a More Convenient and Efficient Rooting Method for Bd21

In tissue culture, the formation of adventitious roots is often a problematic step, and poor adventitious root formation hinders the growth of regenerated shoot and decreases the survival rate [22,23]. Thus, plant regeneration studies often pursue the development of optimum rooting performance. In *Brachypodium*, the regular approach to regenerate adventitious root is culturing on root induction medium (RIM) in the presence or absence of hormones [2,17,24]. However, since hormone-free media generates more well-developed roots compared to hormone-containing RIM [1], the most often used rooting approach is on the CIM-without regulator [1,2,9,16,17,18,25].

To seek the best method for our mature embryo-based system, we compared the efficiencies of root regenerations on hormone-free RIM and RIM with different chemicals (ABT-2 rooting powder (ABT), 3-Indole acetic acid (IAA), Indole-4-butyric acid (IBA), and FPX) (Figure 3b,c). In addition, we also adopted a media-free method called photoautotrophic rooting (PR) in this study (Figure 3a, see details in Methods). A total of 20 calli with 2–3 cm shoots derived from SIM culturing (with 75 μM FPX) were transferred onto different RIM mediums or treated directly using the PR method. After two weeks, root length was measured. For the rooting on RIM, we obtained the average root lengths from 0.83–4.20 cm and, as expected, the root from RIM without hormones showed the longest root size of any of the hormone treatments (Figure 3b,c). Interestingly, rooting with the novel chemical FPX showed a slightly shorter root-size (4.14 cm) than that without hormones (4.20 cm) (Figure 3b,c), suggesting that FPX have a similar rooting ability compared with the hormone-free method.

Strikingly, we observed evidently better-developed roots from the PR treatment with a significantly longer root-size (8.67 cm) than that of all other methods (Figure 3b,c, *p* < 0.01). Moreover, besides the main root, we also observed at least four lateral roots from the PR treatment. In contrast, less than two lateral roots were found from the RIM rooting after two weeks. By examining the survival, we found that the rooting by PR, hormone-free RIM and FPX have a relatively higher survival rate at more than 80% (Appendix A). In contrast, rooting by other hormone-containing RIM has a survival from 45% to 65% (Appendix A). Taken together, the PR showed the best rooting efficiency by generating longer and more vigorous roots than currently used RIM rooting with or without hormones.

### 2.4. Optimization of Conditions for Efficient Transformation from Bd21 Callus 

In our system, embryogenic callus sub-cultured in a long time-range (6–10 weeks, cultured every two weeks to fresh medium) showed yellowish and compact structures, and were transformable (Figure 4b). We then attempted to figure out the optimum embryogenic callus age for our mature embryo system. To this end, calli from 6–10-week-old (300 embryogenic calli from each age) were transformed using the *Agrobacterium* AGL1 with the *pCAMBIA1391* vector (Figure 4a). After two days of co-cultivation with *Agrobacterium*, β-glucuronidase (GUS) activity was analyzed from the transformed calli by GUS staining (GUS transformation efficiency = the number of GUS-positive calli/the number of stained calli × 100%). The results revealed that the TEs of calli increased with age (weeks) and reached the highest value of approximately 99.5% for 9-week old calli (Figure 4b,c). The TE went down to 74.8% for 10-week-old calli. The transformed callus was then grown into seedlings according to the FPX-based shooting and photoautotrophic rooting methods. Positive plantlets were confirmed again by examining the *Hyg* gene by Polymerase Chain Reaction (PCR). We counted the regenerated seedlings to get the final TE (number of positive seedings/number of callus used in transformation) in each callus-age treatment. Interestingly, our data revealed that the young 7-week-old calli showed the highest TE of about 52.6% while the 8-10-week-old calli showed a decreased TE (Figure 4d). These findings may be attributed to the large number of albino plants generated from older calli (G.Y. and K.W. A robust tissue regeneration system in *Brachypodium distachyon*. Unpublished).

## 3. Discussion

Seed dormancy is a very common and polymorphic trait in plants [26]. For the wild grass *Brachypodium distachyon*, seed dormancy has been revealed from the standard lines of Bd21, Bd21-3, and other inbred lines [27]. Results from our study demonstrate that seed dormancy from Bd21 impacts the callus initiation severely (Figure 1). Thus, seeking a simple method that can break seed dormancy in *Brachypodium* is a critical step for the optimization of the mature embryo-based tissue culture system. Strikingly, the easy treatment of cutting the seed can improve callus initiation and growth significantly. Seed dormancy can be released by changing the environmental conditions or treatment with plant hormones, like gibberellin [28]. Besides the unclear knowledge of whether these treatments will improve or inhibit the following callus initiation, all these treatments need certain equipment and long-time processing (from several days to weeks) [28,29]. In contrast, the seed cutting method is simple with no extra treatment and thus has potential for wide usage. In addition, as the great variability for seed dormancy between Bd21, Bd21-3, and other inbred lines [27], it will be of potential interest to test how this method works in other lines that have different seed dormancy levels.

The validation of the new chemical inducer will not only benefit the process of transgenic performance but also add new knowledge to the functional analysis of developmental research in the model grass. FPX, as a new chemical inducer, has been employed in callus and shoot formation in mature and immature embryos in *Brachypodium* [21]. Our results also reinforced the ability of callus and shoot regeneration of FPX in *Brachypodium*. Interestingly, we revealed that FPX has better efficiency for shoot formation than the routinely used KT and TDZ. Since KT is the most often used one in Bd21 in the current immature embryo system [1,2,9,16,17,18], it will be worthwhile to examine if FPX exhibits the superior efficacy compared to other regularly used inducers in various mature- or immature-embryo systems in *Brachypodium*.

PR has been used in the root induction in commercial vegetative propagation since the laborious step of removing media from plantlet roots is not needed again [22,30]. More importantly, practices have demonstrated that PR produces better-developed plantlets with higher survival rates than conventional media-based rooting [31]. The possible explanation for these observations is that the plantlets generated from conventional methods are subjected to excessive water loss upon being transferred from culture vessel to greenhouse and are prone to be damaged during acclimatization [32]. Latter studies on the role of hormones in plant development revealed that the auxins are required to induce roots [23]. More specifically at the first 48–72 h of rooting, auxins release rhizogenic signals to induce the formation of root founder cells, while from ca. 96 h, when the well-recognizable root meristem has been formed and the rhizogenic signal is no longer required [33]. Thus, auxins become inhibitory during the stage of root elongation [33]. Therefore, this may be a reason why rooting on RIM with hormones is inferior to both hormone-free RIM and PR treatment. It is notable that FPX shows a similar rooting ability with hormone-free RIM rooting, suggesting that the type of hormone has a different role in root induction or inhibition. In addition, the poor air permeability of media for RIM rooting should not be ignored as a reason that may cause poor root development. 

The embryogenic calli with a wide range of ages (5–21 weeks, initiated from immature embryos) can be used in *Agrobacterium*-mediated transformation [34]. In our experiment, we also found a large amount of yellow and compact embryogenic calli from the 6–10-week-old group. However, both our study and previous studies have shown that the age of embryogenic calli will severely impact the transformation efficiency in *Brachypodium* [9,34]. The illumination of optimum embryogenic calli will no doubt improve our mature-embryo system to achieve the best TE. As the TE derived from the evaluation based on 300 embryogenetic calli, we believe that a TE about 50% can be achieved by using the optimum 7-week-old calli, and it is comparable with the current TEs (20–70% in Bd21) in both immature and mature embryo systems [15,17,21,35,36].

## 4. Materials and Methods

### 4.1. Plant Material

Seeds of *B. distachyon* inbred line Bd21 harvested once the plants were dry. The seeds within one week after harvesting were used in this study. Cultures were maintained in tissue culture growth room under a 16 h photoperiod and at a temperature of 28 °C and 60% relative humidity.

### 4.2. Callus Induction and Regeneration

Mature seeds were immersed in 75% (*v/v*) ethanol (Sinopharm Chemical Reagent Co., Ltd, Shanghai, China, 80176961) for 30 s followed by 6.25% sodium hypochlorite solution (BBI Life Sciences, Shanghai, China, A501944) for 15 min and rinsed thoroughly three to six times with sterile water. The treated seeds were then put on sterilized filter paper (NEWSTAR, Hangzhou Special Paper Co., Ltd. Hangzhou, Zhejiang, China, Medium102) to absorb excess water.

The procedures and medium formulation for callus and regeneration of Bd21 were according to Alves et al. [15]. The seeds were divided into two groups randomly. One group without any treatment as control, another group of seeds were cut with surgical blades to remove most of the endosperm and named as sheared seeds. Then, 10 seeds from each of the control or sheared treatment were transferred to CIM containing 4.43g/L MS basal medium (Phyto Technology, Shawnee Mission, KS, USA, M519), 30 g/L sucrose (Beijing Solarbio Bioscience &Technology Co., Ltd., Beijing, China, S8270), 2.5 mg/L 2, 4-D, (BBI Life Sciences Corporation, Shanghai, China, A600722), 0.6 mg/L CuSO_4_ (Sinopharm Chemical Reagent Co., Ltd, Shanghai, China, 10008216), and 3 g/L Phytagel (Sigma, Darmstadt, Germany, P8169). For each treatment, three replicates were conducted. The callus was induced for two weeks at 28 ± 2 °C in the dark, and sub-cultured on fresh medium to produce embryogenic callus every two weeks. All calli were transferred without endosperm and germ during the first sub-cultured.

After six weeks, the embryogenic calli were transferred to MS basal medium containing different chemical inducers (0.2 mg/L KT (Sigma, Darmstadt, Germany, K0753), 0.5 mg/L TDZ (Beijing Solarbio Bioscience &Technology Co., Ltd., Beijing, China, T8050) and 75 μM FPX (MedChemExpress, Princeton, NJ, USA, HY-B1124)) for evaluation the efficiency of adventitious shoots formation. Each treatment (20 calli) was conducted with three replicates. 

After shoot elongated into 2–3 cm induced by SIM containing 75 μM FPX, individual shoot was transferred into different rooting mediums with 0.5 mg/L IBA (Beijing Solarbio Bioscience &Technology Co., Ltd., Beijing, China, I8030), 0.6 mg/L IAA (Beijing Solarbio Bioscience &Technology Co., Ltd., Beijing, China, I8020), 45 μM FPX, and 0.05 ppm ABT-2 (ABT Research Center, Chinese Academy of Forestry Sciences, Beijing, China), respectively. Each treatment (20 shoots) was conducted with three replicates for two weeks in the growth chamber at 28 ± 2 °C, with the relative humidity of 60% ± 10%, and 16 h/8 h photoperiod.

### 4.3. Photoautotrophic Rooting 

Twenty calli with 2–3 cm adventitious shoots were washed using 1 mg/L ABT-2 rooting powder solution in Petri dishes to remove the callus, and then the adventitious shoots were immersed into 1 mg/L ABT-2 rooting solution for 3–5 min. The adventitious shoots were then transplanted into the soil mixture (nutrient soil: perlite: vermiculite 2:1:2 (*v/v/v*)) and covered with a transparent plastic cover. Every treatment (20 shoots) was conducted with three replicates. The adventitious shoots grew at 22 ± 2 °C, 60% ± 10% relative humidity, and 16 h/8 h photoperiod for two weeks, during which the adventitious shoots were sprayed with 1 mg/L ABT-2 solution three times a day.

### 4.4. Agrobacterium Strain and Plasmid Vector

*Agrobacterium tumefaciens* strain AGL1 harboring binary vector *pCAMBIA1391* with the GUS reporter gene was used to optimize the transformation system. *A. tumefaciens* strain AGL1 was inoculated into 100 mL LB liquid medium containing 100 mg/L kanamycin (Beijing Solarbio Bioscience &Technology Co., Ltd., Beijing, China, K8020) and 40 mg/L rifampicin (Beijing Solarbio Bioscience &Technology Co., Ltd., Beijing, China, R8010) and grown with shaking conditions (200 r/min) for 20 h at 28 °C. *A. tumefaciens* was collected by centrifugation (3500× *g*, 5 min) and resuspended in MS liquid medium to achieve a final density of OD_600_ = 0.6.

### 4.5. Agrobacterium-Mediated Transformation

After six weeks on the CIM medium, 300 embryogenetic calli were inoculated with *A. tumefaciens* strain AGL1 (OD = 0.6) (harboring binary vector *pCAMBIA1391*) in 50 mL MS liquid medium with 30 μM acetosyringone (AS, Beijing Solarbio Bioscience &Technology Co., Ltd., Beijing, China, A8110) for 15 min and then dried on sterile filter paper. After co-cultured on CIM with one layer of sterile filter paper containing 60 μM AS for 48 h, the calli were then transferred onto the selection medium containing 4.43 g/L MS + 0.6 mg/L CuSO_4_ + 2.5 mg/L 2,4-D + 40 mg/L Hygromycin B (Hyg, Roche, 10843555001) + 300 mg/L timentin (Beijing Solarbio Bioscience &Technology Co., Ltd., Beijing, China, T8660) + 3 g/L Phytagel for 2–3 weeks.

### 4.6. Histochemical Assay of GUS Activity

Histochemical localization was used to assay the GUS gene expression in putatively transformed calli. The calli were treated with GUS staining kit (Real-Times Biotechnology Co., Ltd., Beijing, China, RTU4032) after two days of co-cultivation with *Agrobacterium*. The samples in tissue culture plate 6 (NEST, Corning, NY, USA, TC 703001) were incubated in GUS solution at 37 °C overnight in the dark followed by rinsing in distilled water to remove the staining solution. The calli were then incubated in 70% ethanol and transferred to distilled water. The GUS-positive calli (blue-colored) were photographed with a digital camera [37].

### 4.7. Genomic DNA Extraction and PCR Analysis

Genomic DNA of putatively transformed calli and plant leaves were extracted by the CTAB method [38]. Two pairs of primers were designed and used in the PCR to amplify the GUS (509 bp) and Hyg (527 bp) genes. Primers for GUS sequence are 5′-GCTATACGCCTTTGAAGCC-3′ and 5′-TTGACTGCCTCTTCGCTGTA-3′; for Hyg sequence are 5′- TTCTGCGGGCGATTTGTG-3′ and 5′- AGCGTCTCCGACCTGATG-3′. PCR reaction mixture (20 μL volume) contains 10 μL 2 × Taq Master Mix, 1 μL of forward primer, 1 μL of reverse primer, 1 μL of template DNA, and 7 μL RNase-free water. PCR was performed in a programmable thermal cycler under following conditions: 95 °C for 30 s; 30 cycles of 95 °C for 30 s, 58 °C for 30 s and 72 °C for 1 min; and a final extension cycle of 72 °C for 10 min (Thermo Scientific, Waltham, MA, USA). 

### 4.8. Data Analysis

The SPSS statistical software was used for statistical analysis, and data analysis was performed with the ANOVA method followed by the LSD test to identify significant differences.

## 5. Conclusions

In conclusion, we describe an efficient and mature-embryo derived *Agrobacterium*-mediated transformation system with comparably high TE for the model line Bd21 of *B. distachyon*. We believe these findings will encourage the wide usage of the mature-embryo based system in the model grass *Brachypodium*. In addition, besides the standard line Bd21 and Bd21-3, many inbred lines and wild *Brachypodium* species are developed and subjected to the functional analysis by gene transformation. However, the TE of *Agrobacterium*-mediated transformation in those lines and species is still low or undetermined [2,35,39]. The parameters derived from our system, especially the sheared-seed operation and photoautotrophic rooting will provide potential benefits for the development of an efficient transformation system in those lines or species.

## Figures and Tables

**Figure 1 ijms-20-05448-f001:**
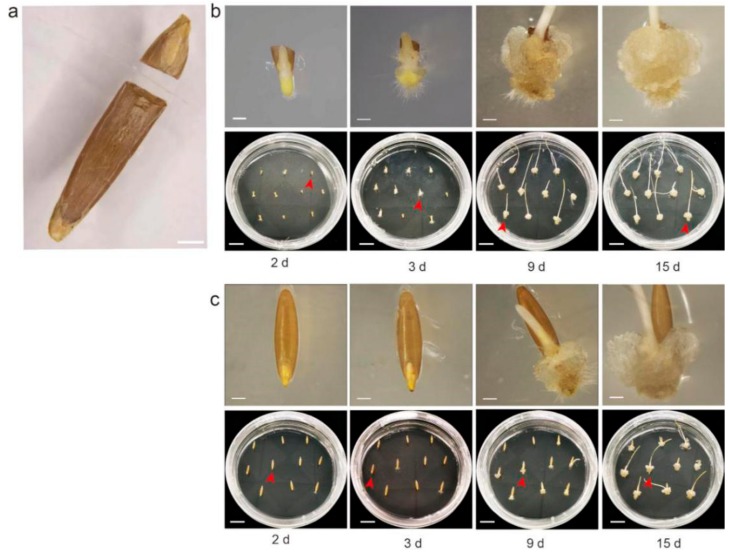
Effect of callus induction from mature embryos in Bd21. (**a**). A photograph depicting a sheared seed, scale bars, 1 mm. (**b** and **c**). The callus induction from sheared seeds (**b**) and untreated seeds (**c**) on different days. The lower panels in (**b**) or (**c**) show the growth of ten seeds on CIM (callus induction medium)_petri dish (MS + 2.5 mg/L 2,4-D + 0.6 mg/L CuSO_4_ + 3 g/L Phytagel), scale bars, 1 cm. The upper panel showed a close-up image of a single seed of the 10 seeds from the corresponding lower petri dish (red arrow), scale bars, 1 mm. For each treatment, three replicates (each of 10 seeds) were conducted.

**Figure 2 ijms-20-05448-f002:**
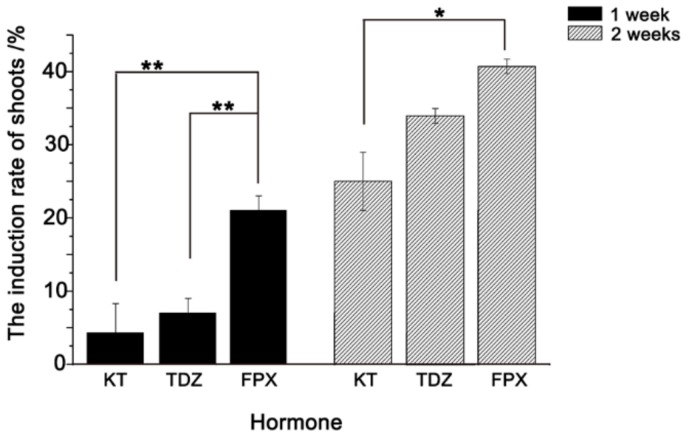
Comparative analysis of shoot induction between KT, TDZ, and FPX. The induction rates of shoots were counted after one and two weeks culturing. The black bar and gray bar represent the induction efficiency of shoots under different hormones after one and two weeks, respectively. For each treatment, three replicates were conducted. The results indicated that significant differences were found between FPX and the other two chemicals of KT and TDZ after one-week growth. After two weeks, significant difference was found between FPX and KT treatments. Error bars represent ± SE, *n* = 3. Asterisk indicates significant difference relative to FPX (Student’s *t* test, * *p* < 0.05 and ** *p* < 0.01).

**Figure 3 ijms-20-05448-f003:**
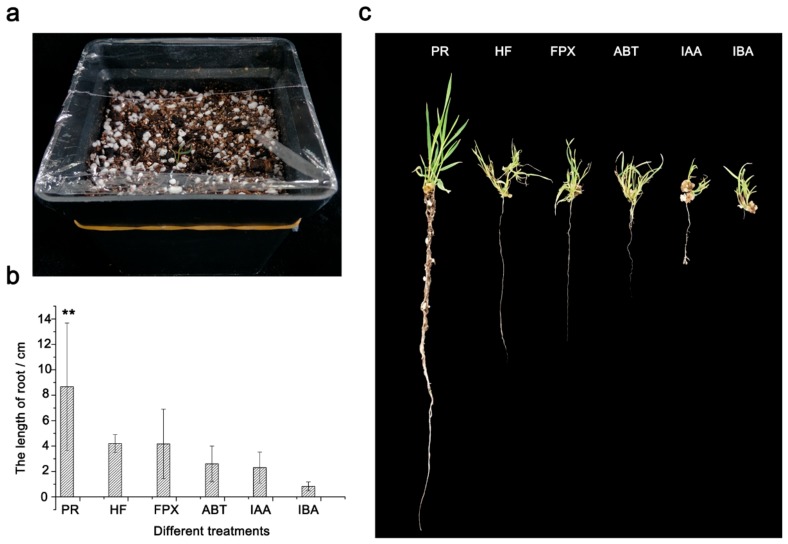
Comparison of the rooting efficiency between PR and RIM-based methods. (**a**). A photograph demonstrating the performance of photoautotrophic rooting. (**b**). The length of regenerated roots cultured on different methods. Root lengths were measured and compared with each other after two weeks of culturing. (**c**). Photos of adventitious roots from different rooting methods. PR, photoautotrophic rooting. HF, hormone-free RIM. ABT, ABT-2 rooting powder (20% NAA and 30% IAA). IAA, 3-Indole acetic acid. IBA, Indole-4-butyric acid. For each treatment, three replicates were conducted. Error bars represent ± SE, *n* = 20. Asterisk indicates significant difference relative to HF (Student’s *t* test, * *p* < 0.05 and ** *p* < 0.01).

**Figure 4 ijms-20-05448-f004:**
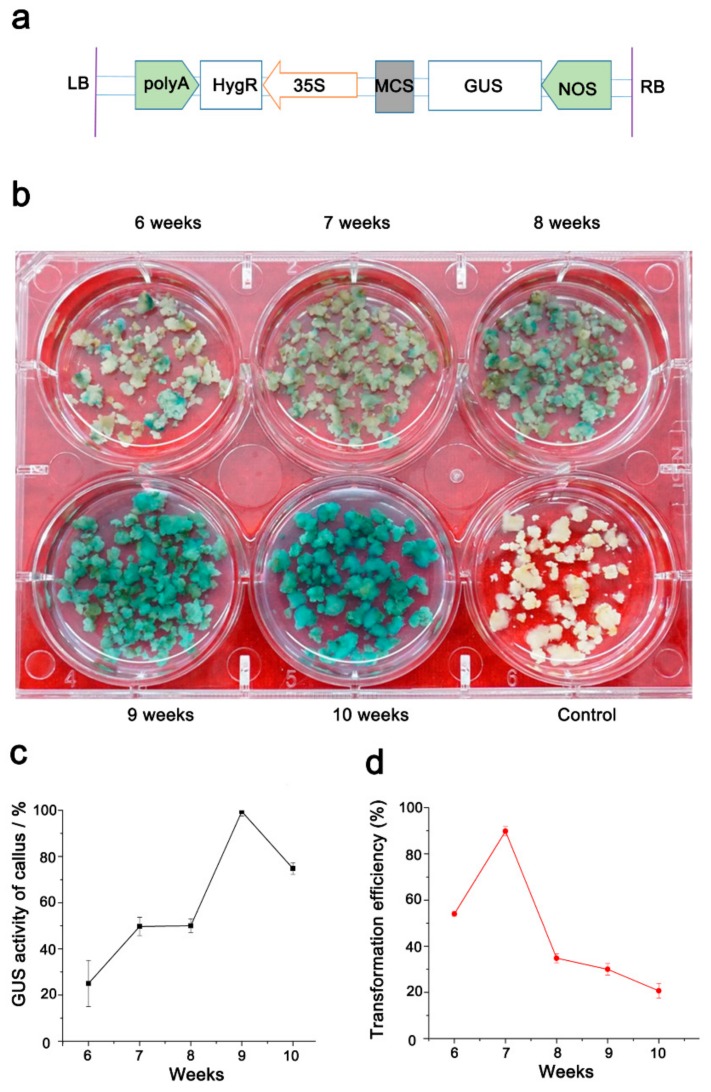
The β-glucuronidase (GUS) activity and transformation efficiency in calli of different ages. (**a**). Schematic diagram of the T-DNA region of the binary vector pCAMBIA1391; LB: left border, RB: right border, polyA-CaMV: polyA signal, HygR: encoding hygromycin B phosphotransferase, 35S: CaMV35S enhance promoter, MCS: multiple cloning site, GUS: encoding β-glucuronidase, NOS-nopaline synthase terminator. (**b**). The GUS activity examination of calli with different ages. (**c** and **d**). Statistical analyses of the GUS activity in callus from different weeks (**c**) and the final transformation efficiency (**d**). For each treatment, three replicates were conducted. Error bars represent ± SE, *n* = 3.

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
