# Peer review of "Optimization of Mature Embryo-Based Tissue Culture and Agrobacterium-Mediated Transformation in Model Grass Brachypodium distachyon"

_ijms, 2019, doi:10.3390/ijms20215448_

Round 1
Reviewer 1 Report
The publication “Optimization of mature embryo-based tissue culture and Agrobacterium-mediated transformation in model grass Brachypodium distachyon” from Yu et al., provides interesting new insights in using mature embryos for callus formation, the chemical inducer Fipexide for callus and shoot formation, and photoautotrophic rooting for improved root regeneration. Overall, those findings will be valuable to the scientific community using Brachypodium distachyon, due to increased transformation efficiency.
Major comments that should be addressed by the authors:
- Please, have the manuscript edited by an English language professional, since there are major inaccuracies that could lead to misunderstanding of the presented results.
- Please, indicate the number of technical and biological replicates, the type of statistical analysis, and the type of error bars presented in the figures.
- Figure 1: Please, add close up images of the calli, especially in panels b and c, ideally, taken under binoculars. Please, also add arrows or asterisks for indication.
- Figure legend of Figure 1: Please, specify single images better, e.g. b-d and f-h and explain.
- For the Fipexide treatments, it would be very interesting to see whether the shoot induction is equally efficient after Agrobacterium transformation, since this is often one of the more problematic steps during tissue culturing.
Since the authors executed the regeneration on FPX plates (explained in paragraph 2.4), it would be simple to add at least those results to the existing data shown.
- Figure 3c likely shows the most striking phenotypes. However, due to the big error bars in Figure 3b, it would be nice to see the variation of phenotypes. Therefore, it would be appreciated if the authors could show other examples for comparisson.
- The authors should consider to write a more detailed Materials and Methods section since this article suggests new methodologies that will be repeated by the plant tissue culturing community. Please, pay attention to details so that the experiments can be repeated as precisely as possible.
- It would be interesting to know how much the age of the mature seeds influence the callus formation capacity, since using 1 week old seeds also represents a time constraint.
- Page 3, line 133-135:”By examining the survival, we found that the rooting by PR, hormone-free RIM and FPX have a relative higher survival rate at more than 80%.” It would be appreciated if those data could be presented in a figure with according statistical tests.
- In paragraph 2.4, please indicate how often and at which time points calli were transferred to fresh medium.
Material and Methods:
- Page 7, line 237-243:”After 6 weeks, the embryogenic calli were transferred to MS basal medium for adventitious shoots formation. Different plant growth regulators (PGRs) were tested such as 0.2 mg/L Kinetin (KT, Sigma, K0753), 0.5 mg/L Thidiazuron (TDZ, Solarbio, T8050), and 75 μM Fipexide (FPX, MEC, HY- B1124) for 2 to 3 weeks. After shoot elongated 2-3 cm in height, individual shoots were placed in different rooting medium with 0.5 mg/L Indole-4-butyric acid (IBA, Solarbio, I8030), 0.6 mg/L 3- Indole acetic acid (IAA, Solarbio, I8020), 45 μM FPX and 0.05 ppm ABT2 rooting powder, respectively.” Please clarify, which shooting treatment was combined with which rooting treatment and include and discuss in the main text.
- Page 8, line 288/289:”…will encourage the wide usage of mature-embryo based systems that has no time limitation…”. Since the authors executed all their experiments with 1 week old seeds, there is a time limitation.
Minor comments that should be addressed by the authors:
- Page 1, line 25: “…ability to induce callus, shoot, and root”, please correct to: callus, shoots, and roots.
- Page 1, line 43:”…but has relatively simple genome”, please correct to: but has a comparatively simple genome”
- Page 2, line 60: “…new regulator chemical Fipexide”, I wouldn’t call Fipexide a regulator, since there is too little scientific evidence which plant components are regulated to induce the observed phenotypes.
- Page 2, line 76/77:”After 15 day growth on CIM (callus will be transfer to new medium for shooting), we obtained active calli from sheared seeds with light yellow color.” Please, clarify if calli are transferred to new media with or without sheared seed attached. The authors should also clarify what is meant by the expression “active calli”.
- Page 2, line 78:”…each callus ball from control seeds showed dark and inactive status.” Please, remove the word ball, and explain what is meant by the expression “inactive status”.
- Page 2, line 78-80:”…callus value from sheared seeds were also highly consistent, while calli from control showed highly diversity in calli value…”. Please, explain what is meant by the word “value”.
- Page 5, line 144/145:”GUS activity was analyzed from the transformed calli by both staining and PCR (GUS gene)”. The presence of the GUS gene does not necessarily represent GUS activity, please re-write.
- Page 6, line 194:”…the formation of root funder cells”, please correct to founder cells.
- Page 6, line 200-202:”In fact, as the requirement and similar performance of rooting in all systems (both immature and mature embryo systems) for tissue culture, we expect that PR has the similar rooting efficiencies.” Please, clarify what is meant by this statement.
- Page 7, line 210:”…evaluation based on 300 embryogenetic caluses”, please, correct to calli.
- Page 7, line 216/217:”Therefore, it is expected that higher TEs are achievable from Bd21-3 when using our mature embryo strategy.” This is pure speculation and does not add anything to the discussion of the presented methods.
- Page 7, line 221:”…harvested within 1week were used…”. Please, clarify the time frame. The seeds were harvested once the plants were dry and within 1 week after harvesting the seeds were used? Or freshly harvested seeds that dried for one week on the plant?
- Page 8, line 269:”…5 days of co-cultivation with Agrobacterium”, on page 5, line 144 it says:”After 2-day incubation”, please clarify that the latter means incubation in GUS solution. In materials and methods, however, it says that the incubation time was between 24-48h, please explain and indicate.
- Page 8, line 271/272:”The GUS-positive calli were photographed with a digital camera “. Please, clarify if Figure 4b shows only GUS positive or the ratio of GUS positive to GUS negative calli.
Author Response
Response to Reviewer 1 Comments
Question 1:Please, have the manuscript edited by an English language professional, since there are major inaccuracies that could lead to misunderstanding of the presented results.
Response:Thank you for your comments. After carefully read this manuscript by all authors, we made some revisions in the manuscript. In addition, a native English speaker also carefully read and edited our manuscript. All the revisions were indicated in the revised manuscript.
Question 2:Please, indicate the number of technical and biological replicates, the type of statistical analysis, and the type of error bars presented in the figures.
Response:Thanks for the valuable suggestion. We revised the figure legends as follows:
Figure 1: “… The lower panels in (b) or (c) show the growth of ten seeds on CIM petri dish ((MS + 2.5 mg/L 2,4-D + 0.6 mg/L CuSO4 + 3 g/L Phytagel), Scale bars, 1 cm. The upper panel showed a close up image of single seed of the ten seeds from corresponding lower petri dish (red arrow). Scale bars, 1 mm. For each treatment, three replicates (each 10 seeds) were conducted.”
Figure 2: “Comparative analysis of shoot induction between KT, TDZ, and FPX. The induction rates of shoots were counted after 1 and 2 weeks culturing. The black bar and gray bar represent the induction efficiency of shoots under different hormones after 1 and 2 weeks, respectively. For each treatment, three replicates were conducted. The results indicated that significant differences were found between FPX and other two chemicals of KT and TDZ after one-week growth. After two weeks, significant difference was found between FPX and KT treatments. Error bars represent ± SE, n=3. Asterisk indicate significant difference relative to FPX (Student’s t test, *P < 0.05 and **P < 0.01).”
Figure 3: “…(c). Photos of adventitious roots from different rooting methods. PR, photoautotrophic rooting. HF, hormone-free RIM. ABT, ABT-2 rooting powder (20% NAA and 30% IAA). IAA, 3-Indole acetic acid. IBA, Indole-4-butyric acid. For each treatment, three replicates were conducted. Error bars represent ± SE, n=20. Asterisk indicates significant difference relative to HF (Student’s t test, *P < 0.05 and **P < 0.01).”
Figure 4: “…For each treatment, three replicates were conducted. Error bars represent ± SE, n=3. ”
In Materials and Methods :
“4.8 Data analysis
The SPSS statistical software was used for statistical analysis, and data analysis performed with ANOVA method followed by LSD test to identify significant differences.”
Question 3:Figure 1: Please, add close up images of the calli, especially in panels b and c, ideally, taken under binoculars. Please, also add arrows or asterisks for indication.
Response:According to the comment, we revised the Figure 1 by adding a more close photo for each stage taken by binoculars and indicated the sample using red arrows.
Question 4:Figure legend of Figure 1: Please, specify single images better, e.g. b-d and f-h and explain.
Response:We revised the figure legends as below, hope it is satisfied:
(b-c ). The callus induction from sheared seeds (b) and untreated seeds (c) on different days. The lower panels in (b) or (c) show the growth of ten seeds on CIM petri dish ((MS + 2.5 mg/L 2,4-D + 0.6 mg/L CuSO4 + 3 g/L Phytagel), Scale bars, 1 cm. The upper panel showed a close up image of single seed of the ten seeds from corresponding lower petri dish (red arrow). Scale bars, 1 mm. For each treatment, three replicates (each 10 seeds) were conducted.
Question 5:For the Fipexide treatments, it would be very interesting to see whether the shoot induction is equally efficient after Agrobacterium transformation, since this is often one of the more problematic steps during tissue culturing.
Since the authors executed the regeneration on FPX plates (explained in paragraph 2.4), it would be simple to add at least those results to the existing data shown.
。
Response:This is really an interesting and constructive suggestion. In fact, we just start another project, in which we will compare the efficient of shoot induction after Agrobacterium transformation for FPX treatment. However, it will take at least three or four month, we thus prefer to not include that in current study.
Question 6:Figure 3c likely shows the most striking phenotypes. However, due to the big error bars in Figure 3b, it would be nice to see the variation of phenotypes. Therefore, it would be appreciated if the authors could show other examples for comparisson.
Response:Thanks for your comments. In fact, we also noticed this problem. A major reason is the sample used in test is only 20 plants which is kind of too few for statistics. However, after carefully searching our in-hand dataset, we find that this is the only photo from this case. You may know, in practice, to reduce the impact on plant growth, we planted the seedling back into soil immediately after measurement. We also not wash the root as the case shown in Figure 3c. So, only one representative plant was selected for each treatment for photo taking. Hope you will understand the situation now.
Question 7:The authors should consider to write a more detailed Materials and Methods section since this article suggests new methodologies that will be repeated by the plant tissue culturing community. Please, pay attention to details so that the experiments can be repeated as precisely as possible.
Response:Thank you for your comments. As suggested, we have rewritten the section of Materials and Methods for providing more details.
Question 8:It would be interesting to know how much the age of the mature seeds influence the callus formation capacity, since using 1 week old seeds also represents a time constraint.
Response:That is true very interesting question. In fact, as mentioned above, we just start a new project, in which we include the study on the question “how much the age of mature seeds influence the callus formation capacity.” We will present the detail result in next paper.
Question 9:Page 3, line 133-135:”By examining the survival, we found that the rooting by PR, hormone-free RIM and FPX have a relative higher survival rate at more than 80%.” It would be appreciated if those data could be presented in a figure with according statistical tests.
Response:We have added the data about survival rate of regenerated plants in Supplementary Figure 1
Question 10:In paragraph 2.4, please indicate how often and at which time points calli were transferred to fresh medium.
Response:We revised regarding sentence as “In our system, embryogenic callus sub-cultured in a long time-range (6-10 weeks, cultured every 2 weeks to fresh medium. ) showed yellowish and compact structures, and were transformable (Figure 4b).”
Question 11:Page 7, line 237-243:”After 6 weeks, the embryogenic calli were transferred to MS basal medium for adventitious shoots formation. Different plant growth regulators (PGRs) were tested such as 0.2 mg/L Kinetin (KT, Sigma, K0753), 0.5 mg/L Thidiazuron (TDZ, Solarbio, T8050), and 75 μM Fipexide (FPX, MEC, HY- B1124) for 2 to 3 weeks. After shoot elongated 2-3 cm in height, individual shoots were placed in different rooting medium with 0.5 mg/L Indole-4-butyric acid (IBA, Solarbio, I8030), 0.6 mg/L 3- Indole acetic acid (IAA, Solarbio, I8020), 45 μM FPX and 0.05 ppm ABT2 rooting powder, respectively.” Please clarify, which shooting treatment was combined with which rooting treatment and include and discuss in the main text.
Response:Thank you for your comments. As suggested, we have revised these sentences as:
Line 121/122:“A total of 20 calli with 2-3 cm shoots derived from SIM culturing (with 75 μM FPX) were transferred onto different RIM mediums or treated directly using PR method. After 2 weeks, root length was measured.”
line 158/160, correct to: “The transformed callus was then grown into seedlings according to the FPX-based shooting and photoautotrophic rooting mentods. Positive plantlets were confirmed again by examining the Hyg gene by PCR.”
Line 253/257, Page 8: “After shoot elongated into 2-3 cm induced by SIM containing 75 μM FPX, individual shoots were transferred into different rooting mediums with 0.5 mg/L IBA (Solarbio, I8030), 0.6 mg/L IAA (Solarbio, I8020), 45 μM FPX and 0.05 ppm ABT-2 (ABT Research Center, Chinese Academy of Forestry Sciences, Beijing, China), respectively.”
Question 12:Page 8, line 288/289:”…will encourage the wide usage of mature-embryo based systems that has no time limitation…”. Since the authors executed all their experiments with 1 week old seeds, there is a time limitation.
Response:The seed dormancy in Brachypodium always appear for the seeds harvested within a certain time. Thus, it is the limitation if conducted regeneration using these new seeds. Our results revealed that cutting seeds can break the dormancy for the new seeds. That is why we say “has no time limitation” here. We have deleted the sentence to avoid confusing meaning in revised manuscript.
Question 13:Page 1, line 25: “…ability to induce callus, shoot, and root”, please correct to: callus, shoots, and roots.
Response:We revised it.
Question 14:Page 1, line 43:”…but has relatively simple genome”, please correct to: but has a comparatively simple genome”
Response: We revised it.
Question 15: Page 2, line 60: “…new regulator chemical Fipexide”, I wouldn’t call Fipexide a regulator, since there is too little scientific evidence which plant components are regulated to induce the observed phenotypes.
Response:We revised as “new chemical inducer Fipexide (FPX)”.
Question 16:Page 2, line 76/77:”After 15 day growth on CIM (callus will be transfer to new medium for shooting), we obtained active calli from sheared seeds with light yellow color.” Please, clarify if calli are transferred to new media with or without sheared seed attached. The authors should also clarify what is meant by the expression “active calli”.
Response:Thank you for your comments. As suggested, we have revised regarding sentences in Materials and Methods section as follow.
“For each treatment, three replicates were conducted. The callus was induced for 2 weeks at 28 ± 2℃ in dark, and sub-cultured on fresh medium to produce embryogenic callus every 2 weeks. All calli were transferred without endosperm and germ during the first sub-cultured. ”
After careful consideration, we deleted the sentence “After 15 day growth on CIM (callus will be transfer to new medium for shooting), we obtained active calli from sheared seeds with light yellow color. In contrast, a large portion of each callus ball from control seeds showed dark and inactive status.”
Question 17:Page 2, line 78:”…each callus ball from control seeds showed dark and inactive status.” Please, remove the word ball, and explain what is meant by the expression “inactive status”.
Response:As mentioned above, we deleted the sentence “After 15 day growth on CIM (callus will be transfer to new medium for shooting), we obtained active calli from sheared seeds with light yellow color. In contrast, a large portion of each callus ball from control seeds showed dark and inactive status.”
Question 18:Page 2, line 78-80:”…callus value from sheared seeds were also highly consistent, while calli from control showed highly diversity in calli value…”. Please, explain what is meant by the word “value”.
Response:It means number and status of callus. We have revised it in the manuscript.
Question 19:Page 5, line 144/145:”GUS activity was analyzed from the transformed calli by both staining and PCR (GUS gene)”. The presence of the GUS gene does not necessarily represent GUS activity, please re-write.
Response: We are sorry for making such confusing, we have revised it as ”GUS activity was analyzed from the transformed calli by GUS staining.”
Question 20:Page 6, line 194:”…the formation of root funder cells”, please correct to founder cells.
Response:We revised it.
Question 21:Page 6, line 200-202:”In fact, as the requirement and similar performance of rooting in all systems (both immature and mature embryo systems) for tissue culture, we expect that PR has the similar rooting efficiencies.” Please, clarify what is meant by this statement.
Simply, we just suggest that we should try the PR method in other labs to reveal if it still has the similar effect in rooting as well as that in our mature embryo-based system. We are sorry for making such a confusing and we have deleted these regarding sentences in the revision.
Question 22:Page 7, line 210:”…evaluation based on 300 embryogenetic caluses”, please, correct to calli.
Response:We revised it.
Question 23:Page 7, line 216/217:”Therefore, it is expected that higher TEs are achievable from Bd21-3 when using our mature embryo strategy.” This is pure speculation and does not add anything to the discussion of the presented methods.
Response:Thank you for your suggestion, we deleted regarding sentences in revised manuscript.
Question 24:Page 7, line 221:”…harvested within 1week were used…”. Please, clarify the time frame. The seeds were harvested once the plants were dry and within 1 week after harvesting the seeds were used? Or freshly harvested seeds that dried for one week on the plant?
Response:It is “within 1 week after harvesting the seeds were used”. We revised it in the manuscript.
Question 25:Page 8, line 269:”…5 days of co-cultivation with Agrobacterium”, on page 5, line 144 it says:”After 2-day incubation”, please clarify that the latter means incubation in GUS solution. In materials and methods, however, it says that the incubation time was between 24-48h, please explain and indicate.
Response:We have revised them as follow:
Lines 144 and 269: “After 2 days of co-cultivation with Agrobacterium” and “…2 days of co-cultivation with Agrobacterium”, respectively.
In materials and methods (line 283): “The samples in tissue culture plate 6 (NEST, TC 703001) were incubated in GUS solution at 37 ℃ for overnight in the dark followed by rinsing in distilled water to remove the staining solution ”
Question 26:Page 8, line 271/272:”The GUS-positive calli were photographed with a digital camera “. Please, clarify if Figure 4b shows only GUS positive or the ratio of GUS positive to GUS negative calli.
Response: Our figure shows the ratio of GUS positive to GUS negative calli.
Reviewer 2 Report
Information about number of seeds for each perti dish and number of replication should be in Material and Methods part. In results Authors could give an information about total number of explants.
The comparison between percentage and number of seeds is quite difficult. Authors should use only % or left the number of seeds (1-2 seeds) and add percentage in the bracket.
In case of Figure 1 and description of callus. I think Authors should add the better pictures of calli, that will be shown the described differences between them.
Fig. 1 In figure legend please add the name of medium used for induction.
Authors compared the FPX and 2,4-D and didn't see any difference. But did You analyzed only the efficiency of callus production? and/or amount of calli? and/or status of calli (inactive or active)?
When Authors use a abbreviation for the first time in manuscript they should always use full name.
Fig. 2. In legend should be KT, TDZ and FPX not FPX, TDZ and FPX. Figure show the precentage of shoot induction so It cannot be described as shoot numbers, only as efficiency. In legend the statistical analysis should be shortly described I mean: * - statistical difference between…. and the same with ** -………. (p<0.01 or 0.05), an information about statistical test.
Fig. 3. In legend please add full name of all abbreviations. **- statistical difference between…. (p<0.01 or 0.05), information about statistical test.
In M&M should be description of statistical analysis.
In M&M (callus induction and regeneration) - Authors mentioned that explants produce an embryogenic callus – but the regeneration is via somatic embryogenesis or shoot organogenesis? In my opinion embryogenic callus is only produce in case of somatic embryogenesis.
ABT2? What is it? Please add full name and shortly described.
Author Response
Response to Reviewer 2 Comments
Question 1:Information about number of seeds for each perti dish and number of replication should be in Material and Methods part. In results Authors could give an information about total number of explants.
Response: We are sorry for making such unclear description. We have made revision by adding details in the materials and methods in revised manuscript. Hope it is satisfied now.
Question 2:The comparison between percentage and number of seeds is quite difficult. Authors should use only % or left the number of seeds (1-2 seeds) and add percentage in the bracket.
Response:Thank you for your comments. As suggested, we have revised it as ” Strikingly, small calli were detected from at least 80% (Callus induction rate = the number of induced calli / total number of seeds ×100%) sheared seeds at day 3 (Figure 1b), when the callus formation had just begun in control seeds (10-20%) (Figure 1c).”
Question 3:In case of Figure 1 and description of callus. I think Authors should add the better pictures of calli, that will be shown the described differences between them.
Response:As suggested, we added new photos of a single seed in Figure 1.
Question 4:Fig. 1 In figure legend please add the name of medium used for induction.
Response:As suggested, we have revised the legend as :“(b-c ). The callus induction from sheared seeds (b) and untreated seeds (c) on different days. The lower panels in (b) or (c) show the growth of ten seeds on CIM petri dish ((MS + 2.5 mg/L 2,4-D + 0.6 mg/L CuSO4 + 3 g/L Phytagel), Scale bars, 1 cm.”
Question 5:Authors compared the FPX and 2,4-D and didn't see any difference. But did You analyzed only the efficiency of callus production? and/or amount of calli? and/or status of calli (inactive or active)?
Response:We revised the regarding sentence as “We also analyzed other comparisons on amount and status of calli. However, we did not obtain any significant difference (data not shown).
Question 6:When Authors use a abbreviation for the first time in manuscript they should always use full name.
Response:We are sorry for making such unprofessional mistakes. We have carefully read the manuscript and made revision according to the suggestion. All the revisions were indicated in revised manuscript.
Question 7:Fig. 2. In legend should be KT, TDZ and FPX not FPX, TDZ and FPX. Figure show the precentage of shoot induction so It cannot be described as shoot numbers, only as efficiency. In legend the statistical analysis should be shortly described I mean: * - statistical difference between…. and the same with ** -………. (p<0.01 or 0.05), an information about statistical test.
Response:We revised the legend according to suggestion as:
Figure 2. Comparative analysis of shoot induction between KT, TDZ, and FPX. The induction rates of shoots were counted after 1 and 2 weeks culturing. The black bar and gray bar represent the induction efficiency of shoots under different hormones after 1 and 2 weeks, respectively. For each treatment, three replicates were conducted. The results indicated that significant differences were found between FPX and other two chemicals of KT and TDZ after one-week growth. After two weeks, significant difference was found between FPX and KT treatments. Error bars represent ± SE, n=3. Asterisk indicate significant difference relative to FPX (Student’s t test, *P < 0.05 and **P < 0.01).
Question 8:Fig. 3. In legend please add full name of all abbreviations. **- statistical difference between…. (p<0.01 or 0.05), information about statistical test.
Response:We revised it as “(c). Photos of adventitious roots from different rooting methods. PR, photoautotrophic rooting. HF, hormone-free RIM. ABT, ABT-2 rooting powder (20% NAA and 30% IAA). IAA, 3-Indole acetic acid. IBA, Indole-4-butyric acid. For each treatment, three replicates were conducted. Error bars represent ± SE, n=20. Asterisk indicates significant difference relative to HF (Student’s t test, *P < 0.05 and **P < 0.01).”
Question 9:In M&M (callus induction and regeneration) - Authors mentioned that explants produce an embryogenic callus – but the regeneration is via somatic embryogenesis or shoot organogenesis? In my opinion embryogenic callus is only produce in case of somatic embryogenesis.
Response:Thank you for your comments. Indeed, plant regeneration occurs in two ways: somatic embryogenesis and organogenesis pathways. We used the somatic embryogenesis. First, we induced the callus from explants, then induced the formation of shoot and root, and eventually forming plants. We are based on the purpose of improving transformation efficiency in the later stage, so we use embryogenic callus. As a mature embryo, the seeds of Brachypodium can produce embryogenic callus through the somatic embryogenesis pathway. So we use this strategy to conduct our experiments.
Question 10:ABT2? What is it? Please add full name and shortly described.
Response:ABT2: ABT2 (This is the full name when asking the company) rooting powder, the main components are 20% NAA and 30% IAA. We have added the information in the figure legend.